# Phytochemical Investigation of New Algerian Lichen Species: *Physcia Mediterranea* Nimis

**DOI:** 10.3390/molecules26041121

**Published:** 2021-02-20

**Authors:** Marwa Kerboua, Monia Ali Ahmed, Nsevolo Samba, Radhia Aitfella-Lahlou, Lucia Silva, Juan F. Boyero, Cesar Raposo, Jesus Miguel Lopez Rodilla

**Affiliations:** 1Laboratory of Vegetal Biology and Environment, Biology Department, Badji Mokhtar University, Annaba 23000, Algeria; kerbouamarwa@gmail.com (M.K.); serradj.monia@gmail.com (M.A.A.); 2Chemistry Department, University of Beira Interior, 6201-001 Covilha, Portugal; nsevolo.samba@ubi.pt (N.S.); ra.aitfella@ubi.pt (R.A.-L.); mlas@ubi.pt (L.S.); 3Department of Clinical Analysis and Public Health, University Kimpa Vita, Uige 77, Angola; 4Fiber Materials and Environmental Technologies (FibEnTech), University of Beira Interior, 6201-001 Covilhã, Portugal; 5Laboratory of Valorisation and Conservation of Biological Resources, Biology Department, Faculty of Sciences, University M’Hamed Bougara, Boumerdes 35000, Algeria; 6Department of Analytical Chemistry, Nutrition and Food Science, Faculty of Chemistry, Chromatographic and mass analysis service (NUCLEUS), University of Salamanca, 37008 Salamanca, Spain; jfbb@usal.es (J.F.B.); raposo@usal.es (C.R.)

**Keywords:** Algerian lichen, *Physcia mediterranea* Nimis, bioactive compounds, n-hexane extract, wax, GC-MS, HPLC-ESI-MS-MS, NMR

## Abstract

The present study provides new data concerning the chemical characterisation of *Physcia mediterranea* Nimis, a rare Mediterranean species belonging to the family Physciaceae. The phytochemical screening was carried out using GC-MS, HPLC-ESI-MS-MS, and NMR techniques. Hot extraction of n-hexane was carried out, followed by separation of the part insoluble in methanol: wax (WA-hex), from the part soluble in methanol (ME-hex). GC-MS analysis of the ME-hex part revealed the presence of methylbenzoic acids such as sparassol and atraric acid and a diterpene with a kaurene skeleton which has never been detected before in lichen species. Out of all the compounds identified by HPLC-ESI-MS-MS, sixteen compounds are common between WA-hex and ME-hex. Most are aliphatic fatty acids, phenolic compounds and depsides. The wax part is characterised by the presence of atranorin, a depside of high biological value. Proton 1H and carbon 13C NMR have confirmed its identification. Atranol, chloroatranol (depsides compound), Ffukinanolide (sesquiterpene lactones), leprolomin (diphenyl ether), muronic acid (triterpenes), and ursolic acid (triterpenes) have also been identified in ME-hex. The results suggested that *Physcia mediterranea* Nimis is a valuable source of bioactive compounds that could be useful for several applications as functional foods, cosmetics, and pharmaceuticals.

## 1. Introduction

From the Greek word, “leikhen” lichen was first used to designate a plant in the 4th century BC by Theophraste [1]. This small organism has been integrated into the fungal kingdom and results from the symbiotic association of a fungus called mycobiont and a green alga or a cyanobacterium called photobiont [2,3]. An estimated more than 17,000 species of lichen exist today, extending from the tropics to the polar regions [4]. The symbiosis gives lichens a specific structure and reproduction to each constituent alone. Unlike higher plants, they have no root, stem, or leaf, but a rudimentary vegetative apparatus: the thallus [5]. Moreover, they grow on a wide variety of substrates including soil, bark, bare rock surfaces, leaves of vascular plants, barnacle shells, and other lichens [4].

The mycobiont plays the most determining role in the morphology and structure of the lichen [6]. The thallus carries the elements necessary for reproduction. A great diversity of shapes and colours defines seven main types of lichens [7]. These organisms tolerate large temperature variations, extreme desiccation, and intense exposure to ultraviolet [8]. In addition, this tolerance to light and desiccation is better in lichen compared to isolated partners [9]. As a two or even multiple partnerships, lichens can respond more sensitive to different environmental signals in complex ecosystems than algae and fungi growing unprotected and living aposymbiotically according to Reyes et al., 1996 [10]. 

Lichens produce many unusual secondary metabolites that have not been discovered in other plants [11]. The uniqueness of many lichen substances attracted the first chemists’ attention in the mid-19th century. Before this, they were used as dyes for textiles and as additives for soap and perfume making, and a considerable number for healing diseases [12,13]. Lichens produce more than 800 potentially bioactive compounds [14,15,16]. Among these compounds are nitrogen, phosphorus and sulfur, polyols, carbohydrates, aliphatic and cycloaliphatic compounds, aromatic compounds, meta- and para-depsides, and depsidones, dibenzofurans, diphenylethers, naphtopyrans, biphenyls, diphenylmethanes, nostoclides, xanthones, quinones, naphthoquinones, and usnic acid. Esters, terpenes, steroids, terphenylquinones, and pulvinic acid derivatives also occur [16,17,18]. The majority of lichenic substances are small aromatic polyketides. They are synthesised during the mutualistic relationship (symbiosis) of lichens with green photobionts [13,19]. These substances are often absent in cyanobacterial lichens replaced by other interesting secondary metabolites, e.g., terpenes and terpenoids [20,21].

Recently, lichens have attracted several researchers worldwide to review their therapeutic and cosmetic value in traditional medicine. A wide range of species have revealed effective biological activities such as antioxidant, cardiovascular protectors, antimicrobial, anti-inflammatory, analgesic, antipyretic, antiviral, anti-insecticidal, antibiotic, antifungal, antidiabetic, and anticancer [22,23,24,25,26,27,28,29,30,31]. Various applications of lichen extracts as a treatment for skin conditions, wounds, respiratory and digestive problems, as well as for obstetric and gynaecological problems have also been recorded [32]. The pharmacological potential of lichens is due to the presence of unique secondary metabolites which subsequently became excessively attractive to the pharmaceutical and cosmetic industries. As a result, a multitude of optimised processes for the quantitative extraction of these lichen metabolites has been developed [33].

In Algeria, several investigations have been carried out to explore their diversity [34,35,36,37,38,39,40,41,42,43,44,45]. More than 1085 species of lichens have been identified, 64 of which are endemic [41]. Most research has been focused on their use as bioindicators of air pollution [42,43,46,47,48,49]. However, many regions remain unexplored, and research reports on lichen chemistry in this richly diverse country are very limited.

In the present investigation, we analysed the phytochemical profile of *Physcia mediterranea* Nimis (Figure 1), belonging to the family of Physicaceae. The lichen, a new species recently identified in Algeria, was collected in the El-Kala National Park, a remarkable and culturally rich region with more than 117 species of lichens identified. Several species of this area, both plant and animal, are protected in Algeria and belong to the IUCN (International Union for Conservation of Nature) red list. Indeed, the National park of El-Kala was classified as “Reserve of the biosphere”, by UNESCO, on 17 December 1990 [50]. It can be concluded that the Algerian species of lichens that have been little or not studied deserve special attention with research involving both the chemical part and the biological one. This would allow us to increase the contribution to discovering new compounds that may serve as models for new drugs with therapeutic properties. The literature search reveals that still no studies have been done on *Physcia mediterranea* Nimis species’ products. Therefore, our work aims to isolate and investigate the chemical constituents in different organic lichen extracts from *Physcia mediterranea* Nimis using GC-MS, followed by HPLC-ESI-MS-MS and NMR analysis.

## 2. Results and Discussion

### 2.1. GC-MS Analysis

The GC-MS analysis of ME-hex extract is shown in Figure 2. The analysis allowed us to detect ten components. Nine products were identified by comparing their mass spectra with reference spectra from the NIST and Wiley databases (Table 1). The major chemical compounds identified in ME-hex from *Physcia mediterranea* included derivatives of methylbenzoic acids (56%) and free fatty acids (36.1%). A diterpene with a Kauran skeleton ((−)-ent-Kauran-16a-ol) (Figure 3) was also identified (3.8%) [18]. Only 2.2% of the entire extract was not identified.

Derivatives of methylbenzoic acids are commonly present in different genera of lichen-like *Stereocaulon halei* [51], *Parmotrema mesotropum* [52], *Cassipourea malosana* [53], *Cetraria islandica* [54], *Usnea longissima* [55,56], *Stereocaulon paschale* [57], and *Parmelia sulcata* Taylor [58]. Atraric acid (Figure 2), widely present in some species such as *Hypogymnia physodes*, *Evernia prunastri*, and *Parmelia sulcata*, growing on the same host tree (*Prunus domestica*) [59], can be in the free or complexed form and serves as a basis for the composition of depsides and depsidones [60]. It is considered a specific antagonist of androgen receptors and therefore inhibits human prostate cancer growth [60,61]. Atraric acid shows nematocidal, antioxidant, antimicrobial, and anti-inflammatory properties in vitro, and inhibits carrageenan-induced oedema and wound healing activity in vivo [62,63].

Lichens contain many of the fatty acids commonly found in higher plants [64]. Indeed, many common lichen genera include species with multiple strains of fatty acids [65,66,67]. These lipid profiles, most often treated as chemotypes, have been used by many researchers to taxonomically classify certain lichens, such as Cladonia [68], Lepraria [69], Parmelia [70], Tephromela s.lat [71], and Mycoblastus sanguinarius [72]. Although they are widely present in some genera, fatty acids were of no taxonomic importance and were omitted during the first chemical studies on lichens [72,73,74]. By chemical and biochemical comparisons, a mechanistic relationship between polyketide and fatty acid biosynthesis has been recognised. The carbon backbones of the molecules are assembled by successive condensation of acyl units [75]. Another important fatty acid role was cell signal transduction [76] as well as chemical protection. Therefore, they allow the lichens to survive as environmental conditions change [8,9,10].

In our study, palmitic acid is present in high concentration (24%) compared to oleic (4.2%), stearic acid (3.8%), and linoleic (3.2%) acid (Table 1 and Figure 2). Various works using different growing conditions explain the variations in fatty acid in lichens [77,78]. Molina et al. (2003) studied the lichen *Physconia distorta* and suggested a close relationship between the synthesis of secondary metabolites and fatty acid metabolism. Mycobiota grown in a glucose-enriched medium favoured the production of fatty acids [79]. Another important factor that could influence the production of fatty acids in lichens is temperature. According to several studies, the degree of unsaturation varies with the season and decreases with increasing temperature [80,81]. In the thallus of *Teloschistes flavicans*, the saturated fatty acids, palmitic and stearic, were more abundant in February. In contrast, in August, when the average temperature was 23 °C, there was an increase in oleic and linoleic fatty acids [82].

In addition to the stress due to the decrease in temperature, nitrogen deprivation and light intensity are also known to promote fatty acid accumulation [80,82,83]. However, these factors are not necessarily the unique parameters that determine the fatty acid content, but rather genetics, combined with environmental conditions (e.g., altitude, air pollution, seasonal effects), must also be taken into account [67,84].

### 2.2. HPLC-ESI-MS-MS Analysis

#### 2.2.1. *WA-hex* Extract Analysis of *Physcia mediterranea*

Forty-five peaks (Appendix A) were detected for the first time in a WA-hex (Table 2) using LC/ESI/MS/MS in negative mode. Identifying major compounds was simplified by interpreting their MS/MS spectra, provided in our system resource and comparison with the literature. The 37 compounds identified were mainly paraconic and aliphatic acids, depsides (aromatic polyketides), phenolic compounds, and diterpenes. Only eight compounds could not be identified. The representative chemical structures are presented in Appendix A.

Twenty-nine paraconic and aliphatic acids were identified: peaks 1–2, 5–7, 9, 13–19, 21–24, 26–29, 31, 35, 37, 39, 40, 42, and 44 using UHPLC-ESI-MS-MS analysis [13,18]. Among these compounds is fumaric acid (Appendix A), a valuable compound used in food, beverages, detergents, animal feed, pharmaceuticals, and various industrial products [85,86,87,88]. Similarly, for traumatic acid, a phytohormone belongs to the class of cytokinins, a study has demonstrated its positive influence on oxidative stress parameters in normal human fibroblasts [89]. It is also effective against breast cancer cells and has potential anticancer properties and tumour prevention activity. Traumatic acid leads to decreased cell proliferation and viability, GSH/GSSG ratio, and thiol group content. It increased caspase activity, membrane lipid peroxidation, and ROS content, simultaneously reducing breast cancer cell growth through the influence of oxidative stress on apoptosis [90].

Three phenolic compounds were identified and assigned to peaks 4, 8, and 10 using UHPLC-ESI-MS-MS. Peak 4 and peak 10 were identified as *p*-Coumaric acid 6,7-Dihydroxycoumarin (Esculetin) (Appendix A) whose molecular anions were at *m*/*z* 163.0395 and 177.0186, respectively. To our knowledge, this is the first time that compounds such as these have been found in lichen material. They possess diverse biological and pharmacological properties, including anti-asthma, anti-inflammatory, anti-nociceptive, antioxidative, antitumor, and antiviral activities [91,92,93,94,95,96,97]. Gingerol, a phenolic compound (Appendix A), has been identified in the extract WA-hex. Its peak 8 would correspond to [M − H]^−^ ion at 293.1762 (Table 2).

Peak 11 and 12 were identified as depsides: atranorin ([M − H]^−^ ion at *m*/*z* 373.0929) and chloroatranorin ([M − H]^−^ ion at *m*/*z* 407.0539), respectively (Table 2, Appendix A). Atranorin, a derivative of β-orcinol, is one of the most common secondary metabolites of lichen [98]. It is particularly present in the lichen families Cladoniaceae, Lecanoraceae, Parmeliaceae, and Streocaulaceae [99,100,101]. In the Physcia genera, there are *Physcia caesia* [102], *Physcia aipolia* [103,104], *Physcia alnophila* [105], and *Physcia sorediosa* [106]. In recent years, atranorin has been the most extensively studied. Indeed, it has shown antioxidant [107], antimicrobial [108], anti-inflammatory [109], antinociceptive [110], wound healing [111], and photoprotective properties [112]. Additionally, it exerted strong inhibitory effects on cancer cell proliferation, migration, and actin cytoskeleton organisation [113].

Portentol, peak 32 ([M − H]^−^ ion at *m*/*z* 309.1743) and Stephanol peak 36 ([M − H]− ion at *m*/*z* 397.2266), which are cycloaliphatic compounds were also identified in WA-hex (Table 2, Appendix A). Likewise for asebotoxin I, a toxic diterpene has been identified at the peak 43 ([M − H]^−^ ion at *m*/*z* 425.2581). It was initially and only discovered in the plant *Pieris japonica* [114], but our results show the opposite, i.e., that the compound can be detected in lichen species.

#### 2.2.2. ME-hex Extract Analysis of *Physcia mediterranea*

In the present study, the analysis of the phytochemical profile of ME-hex using UHPLC-ESI-MS/MS, in negative ion mode, resulted in the detection of 54 significant compounds indicated in Appendix A. Only four compounds could not be identified. The identified compounds are of paraconic and aliphatic acids, aromatic polyketides (depsides, depsones, and phenyl ethers), phenolic acids, sesquiterpenes lactones, triterpenes, carboxybenzaldehyde, and carboxyphthalide types (Table 3).

Thirty-seven paraconic and aliphatic acids corresponding to peaks 1–3, 5, 6, 8–10, 12, 14–19, 22, 24–30, 38, 39, and 41–52 were identified using UHPLC/ESI/MS/MS analysis. 3,5-Dimethoxyciclohexanecarboxylic acid (Figure 4) was identified as peak 1 (molecular anion at *m*/*z* 188.1043). The fragmentation of peak 1 produced ion at *m*/*z* 143.8651 [M − H − CO_2_]^−^, and 141.8670 [M − H − CO_2_]^−^. Peak 2, with an [M − H]^−^ ion at *m*/*z* 294.0741, was identified as 6-(hydroxymethyl)-3,5-bis(methoxycarbonyl)-2,4-dimethylcyclohex-1-ene-1-carboxylic acid (Figure 4). The fragmentation of peak 1 produced ion at *m*/*z* 249.0771 [M − H − CO_2_]^−^, 234.0530 [M – H − CO_2_ − CH_3_]^−^, and 207.0695 [M − H − CO_2_ − CH_3_]^−^, confirming this compound. Peak 3 was identified as 3,5,6-hydroxymethyl-2,4-dimethylcyclohex-1-ene-1-carboxylic acid (Figure 4), which showed a [M − H]^−^ at *m*/*z* 244.1311. Major diagnostic daughter MS ions of the compound were [M − H − CO−CH_3_OH]^−^ and [M − H − H_2_O_2_]^−^ (183.1025 and 176.6474 a.m.u, respectively). Peak 5 with a [M − H]^−^ ion at *m*/*z* 226.1201 was identified as 5-formyl-3-hydroxymethyl-2,4,6-trimethylcyclohex-1-ene-1-carboxylic acid (Figure 4). The fragmentation of the peak produced ions at *m*/*z* 207.1023 and 97.0284 confirming this structure. Peak 6 was identified as 3,5-dihydroxy-2,4,6-trimethylciclohexenecarboxilic acid (molecular anion at *m*/*z* 200.1046) (Figure 4). Major daughter ions of peak 6 were at *m*/*z* 183.4492, 162.8385, and 114.9508. Peak 8 with a [M − H −]^−^ ion at *m*/*z* 242.1153 was identified as 4-O-demethylbaeomycesic acid, and their major diagnostic daughter MS ions were [M − H − CO_2_]^−^ and [M − H − 2CH_3_OH]^−^, (218.8167 and 172.6485 a.m.u, respectively). The 2,4-dihydroxy-3,5,6-trimethylcyclohexane-1-carboxylic acid (Figure 4) was identified as Peak 8 ([M − H]^−^ at *m*/*z* 202.1202). The fragmentation of this compound produced three ions at *m*/*z* 197.6263, 164.8350 and 139.1124 confirming its structure. Peak 12 was identified as 4-hydroxy-2,5-dimethylcyclohex-1-ene-1-carboxylic acid (Figure 4), which showed a [M− H ]^−^ peak at *m*/*z* 170.0936. Major diagnostic daughter MS ions were [M − H − CO]^−^, [M − H − CO_2_]^−^, and [M − H − CO_2_]^−^ (193.0514, 140.9478, 124.9796, and 104.4180 a.m.u., respectively). Peak 14 was assigned to 6-(1-Oxopentyl)-1-cyclohexene-1-carboxylic acid (Figure 4) on its resolution molecular anion at *m*/*z* 210.1253 ([M − H]^−^ peak). It produced major diagnostic MS ions at 164.8947, 146.9850, and 105.0336.

According to the results (Table 3), four depsides compounds were identified corresponding to peak 4, 11, 23, and 35 in the ME-hex extract. Peak 4 was identified as atranol (Figure 5) (molecular anion at *m*/*z* 152.0465). The fragmentation of peak 4 also produced ions at 123.0444 [M − H − CO_2_]^−^, 105.0335 [M − H − CO_2_− H_2_O_2_]^−^, and 81.0335 [M − H − CO_2_ − H_2_O_2_]^−^ confirming this compound. Chloroatranol and 8-Hydroxydiffractaic acid identified at peak 11 and 23, showed [M − H]^−^ ions at *m*/*z* 186.0079 and 390.1315, respectively. This is also chloratranorin (Figure 5) at peak 35, which shows [M − H]^−^ ions at *m*/*z* 408.0611.

A despsone named allo-protolichesterinic acid was also detected in ME-hex extract. It would correspond to the peak 21 and shows [M − H]^−^ ions at *m*/*z* 324.2303. Moreover, two phenolic acids were identified, corresponding to peak 13 and 31: p-Coumaric acid and dihydroxycoumarin (Figure 5). They showed [M − H]^−^ ions at *m*/*z* 164.0470 and 178.0259 respectively. Peak 34 was identified as 7-chloro-3-oxo-1,3-dihydroisobenzofuran-5-carboxylic acid (carboxyphthalide) (Figure 5), which showed a [M − H]^−^ peak at *m*/*z* 211.9873. It produced major diagnostic MS ions at *m*/*z* 138.9948 and 103.0180, confirming this compound.

It is similar for the compounds fukinanolide (sesquiterpene lactones) at peak 20 and leprolomin (diphenyl ether) (Figure 5) at peak 37. They showed a [M − H]^−^ peak at *m*/*z* 312.2302 and 366.2403 respectively. In the study by Zhang et al. (2016), fukinanolide, also called bakkenolide A, extracted from the plant *Petasites tricholobus*, showed anti-inflammatory properties in the treatment of leukaemia [115]. In the lichens group, the sesquiterpene has only been identified in *Cetraria islandica* [116].

Finally, two triterpenes, muronic acid at peak 32 and ursolic acid at peak 39, were also identified (Figure 5) and produced major diagnostic MS ions at *m*/*z* 390.1314, 456.3605. Muronic acid was previously identified in *Usnea rubicunda, Usnea subfloridana* [117], and *Punctelia microsticta* [118]. On the other hand, ursolic acid (3β-hydroxy-urs-12-ene-28-oic acid) is widespread in the vegetable kingdom [119]. The lichenic species *Ramalina hierrensis* [16], *Ramalina hierrensis* [120], *Stereocaulon evolutum* [121], and *Pannaria tavaresii* [122] also contain this compound.

### 2.3. NMR Analysis

1H and C13 NMR analysed the two samples from WA-hex and ME-hex. The results we obtained reveal the predominance of the secondary metabolite atranorin only in the WA-hex sample. The structure has been characterised, and the NMR spectra are compared with those of a previously isolated sample of *Physcia sorediosa* [106,123], demonstrated in Appendix A.

Atranorin (Figure 6) is the most common secondary metabolite in lichens and is mainly found on lichens’ surface (cortex) [124]. It acts as a photo-buffer because it reflects harmful UV rays to the thallus’s surface and allows the lichens to live in areas receiving intense solar radiation [125].

Several factors can influence atranorin concentrations in lichens. They fluctuate with the seasons [126,127] and vary according to the habitat [128]. The method of preparation and extraction of lichens can also influence the concentration of this metabolite. Conventional organic solvents (such as hexane and acetone) are commonly used for its extraction as hydrophobic [129]. Our study used n-hexane with Soxhlet extraction method, which is of choice in studying organic analytes extracted from lichens [130]. It is still used to date to extract organic air pollutants, organochlorinated pesticides and insecticides from the lichen matrix [131,132].

According to Komaty et al. (2015), the lichen grinding method can affect the extraction efficiency and even be used to selectively increase the extraction efficiency of certain secondary metabolites such as atranorin [133]. To achieve a higher yield, use a blender instead of a ball mill, as it selectively grinds the cortex into a fine powder, which can be recovered from the larger medulla pieces. A ball mill or a mortar and pestle technique will reduce the whole lichen to powder, which will reduce the extraction efficiency of the atranorin [129]. *Pseudevernia furfuracea* is a lichen widely used as a raw material in the perfume and cosmetics industries due to its richness in aromatic compounds [33,134,135,136,137]. Microwave-assisted extraction of this lichen has increased atranorin extraction efficiency by a factor of five [133].

## 3. Materials and Methods

### 3.1. Lichen Material

The saxicolous lichen specimen *Physcia mediterranea* Nimis was collected at Ain Tebib (Sector Oum tboul) on the rock, at an altitude of 120 m above sea level, coordinate 36°49′ 09″ N; 08°31′ 33″ E in June 2017 (Figure 7). Ain Tebib station is located in the national park of El Kala (80,000 ha). The collected lichen samples were packed in polyethene bags and stored at 4 °C until further processed. Professor Monia Ali Ahmed has identified *Physcia mediterranea* Nimis (Figure 1)*,* lichenologist and research director of the Pathology of Ecosystems team at the University of Badji-Mokhtar, Annaba, Algeria. Botanical description of *Physcia mediterranea* Nimis is in Appendix A. The identification was confirmed by Pr Jean Michel Sussey, lichenologist at the French Association of Lichenology (AFL). This sample has been deposited in Badji-Mokhtar University, Annaba, code AAM-1.

### 3.2. Sample Preparation

The lichens were washed with tap water to remove the dust and other foreign materials. The washed samples were dried under shade for a week. The lichen was ground using a grinder. The preparations were then pulverised into powdered form by using heavy-duty blender.

### 3.3. Preparation of Physcia mediterranea Extracts

The powder samples (24 g) of *Physcia mediterranea* Nimis were extracted with the solvent n-hexane (500 mL) using a Soxhlet extractor 24 h. After complete extraction, the solvent was evaporated using a rotary evaporator under reduced pressure to obtain n-hexane extract (1.026 g). It was then extracted with hot methanol (60 °C) to obtain two parts: insoluble precipitate representing lichen wax (WA-hex) and the methanol soluble part (ME-hex). Both extracts (WA-hex/ME-hex) were completely evaporated using a rotary evaporator under reduced pressure to obtain dry extracts (0.552 g/0.300 g), respectively.

### 3.4. Instrumentation and Analysis Parameters

In this present study, the chemical composition of the two fractions WA-hex and ME-hex of Algerian *Physcia mediterranea* Nimis was analysed using the HPLC-ESI-MS-MS method. Moreover, we combined GC-MS’s ability with the targeted metabolomics of HPLC-ESI-MS-MS methods to characterise the composition of n-hexane extract (ME-hex) for the first time. In addition, most compounds potentially present in the wax (WA-hex) are detected and characterised using the NMR method.

#### 3.4.1. GC-MS Analysis

For the GC-MS analysis, an Agilent MS220 (Varian, Inc. Walnut Creek, CA, USA) mass spectrometer coupled to a 7890A GC. The oven temperature was initially set to 50 °C, held for 5 min, then a ramp of 30 °C/min was applied up to 270 °C that was held for an additional5 min. MS spectra were acquired in EI mode with a mass range from 50 to 600 a.m.u. Before being injected into the GC-MS system, the ME-hex fraction was pre-esterified with diazomethane, in order to identify eventual less polar compounds in this fraction. It was then solubilised in dichloromethane and injected into the apparatus.

#### 3.4.2. NMR Analysis

Proton 1H and carbon 13C NMR spectroscopy were recorded on a Brüker Advance III 400 MHz spectrometer (Brüker Scientific Inc, Billerica, MA, USA) at 400 MHz for proton and 100 MHz for carbon. The recovered WA was dissolved in deuterated solvent’s (CDCl_3_) (5 mg/mL), at room temperature. The solution was transferred to 5 mm outside diameter tubes, and the spectra were acquired at room temperature. The deuterated solvent’s residual peak signal was for 1H spectra at 7.26 ppm and 13C spectra at 77.2 ppm. The chemical deviations (δ) are expressed in parts per million (ppm) and the coupling constants (J) in Hertz. The data was processed using TOPSPIN 3.5 software (Brüker Scientific Inc.).

#### 3.4.3. HPLC-ESI-MS-MS

The method was carried out on an orbitrap Thermo q-Exactive mass spectrometer coupled to a Vanquish HPLC. A Kinetex XB-C18 (Phenomenex) with a particle size of 2.6 microns, 100 mm of length, and a diameter of 2.1 mm was used as a column. The mobile phases were 0.1% formic aqueous solution (A) and, acetonitrile (B). The gradient program (time (min), % B) was: (0.00, 50); (20.00, 100); (25.00, 100); (26.00, 50). The flow rate was 0.200 mL min^−1^ and the injection volume was 10 µL.

The ionisation electrospray in negative mode was used. The following analysis parameters were: electrospray voltage −3.8 kV, sheath gas flow rate, 30; auxiliary gas unit flow rate, 10; drying gas temperature, 310 °C; capillary temperature, 320 °C; S-lens and RF level, 55. The acquisition was performed in a mass range from 100 to 1000 a.m.u. An auto MS2 program was used with a fragmentation voltage of 30.

## 4. Conclusions

Knowledge of the chemical constituents of lichens is invaluable as this information will be useful for the synthesis of potential new chemical substances. Many researchers report such phytochemical screening of various lichens [103,106,116,117,123,124]. A growing body of evidence indicates that lichens’ secondary metabolites play an essential role in human health and may be nutritionally important [24,26,27,29,30,31,58,112,118,119,120]. In the present study, we have identified and chemically characterised Algerian *Physcia mediterranea* Nimis for the first time. The extraction of these metabolites was carried out with hot hexane. A methanol-crystallisation process allowed us to characterise and identify the atranorine depside, as a major component of lichen wax. In this work, several aromatic acids, a kaurane, and fatty acids have been identified by GC-MS of ME-hex. The UHPLC-ESI-MS-MS technique has been used to analyse the crystallised fraction, WA-hex, whose major product is atranorin and chloroatranorin with the minority products identified. In ME-hex fraction, this technique identifies paraconic and aliphatic acids, depsides, depsones, phenolic compounds, sesquiterpenes, triterpenes, and phenyl ethers in addition to other minority derivatives. This study reveals for the first time the different compounds of *Physcia medeterranea* considered as a rare international species; furthermore, it highlights the importance of the lichens of Algeria as a promising source of bioactive molecules.

## Figures and Tables

**Figure 1 molecules-26-01121-f001:**
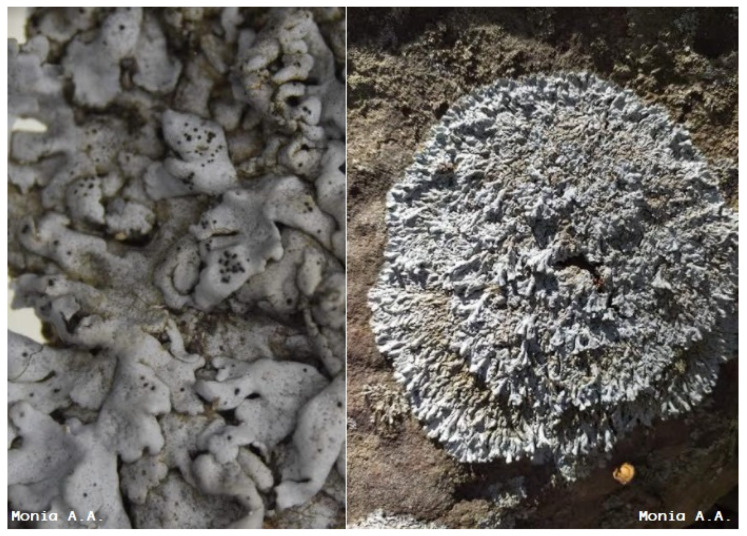
Algerian *Physcia mediterranea* Nimis photographed in the national park of El Kala, Algeria.

**Figure 2 molecules-26-01121-f002:**
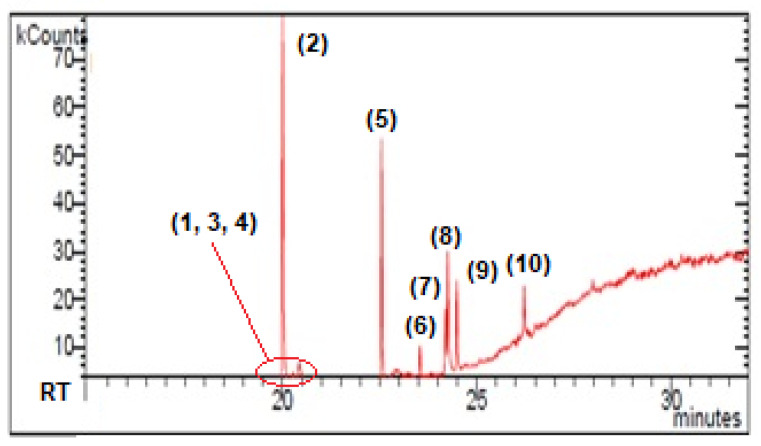
GC-MS chromatogram of *Physcia mediterranea* (ME-hex part). RT = retention time. (1), (2), (3), (4) = derivatives of methylbenzoic acids; (5), (6), (7), (8) = fatty acids; (9) = diterpene; (10) = unknown.

**Figure 3 molecules-26-01121-f003:**
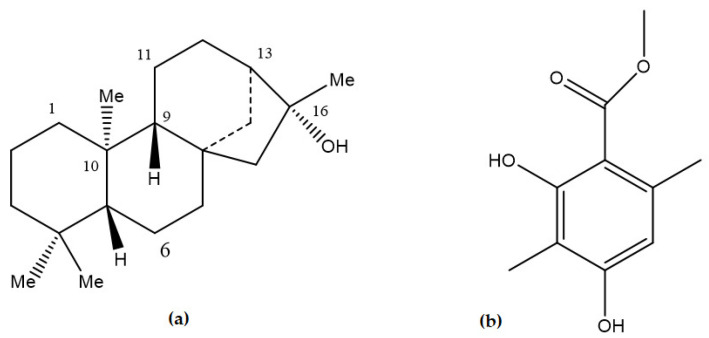
Chemical structure of (**a**) (−)-ent-Kauran-16a-ol and (**b**) Atraric acid.

**Figure 4 molecules-26-01121-f004:**
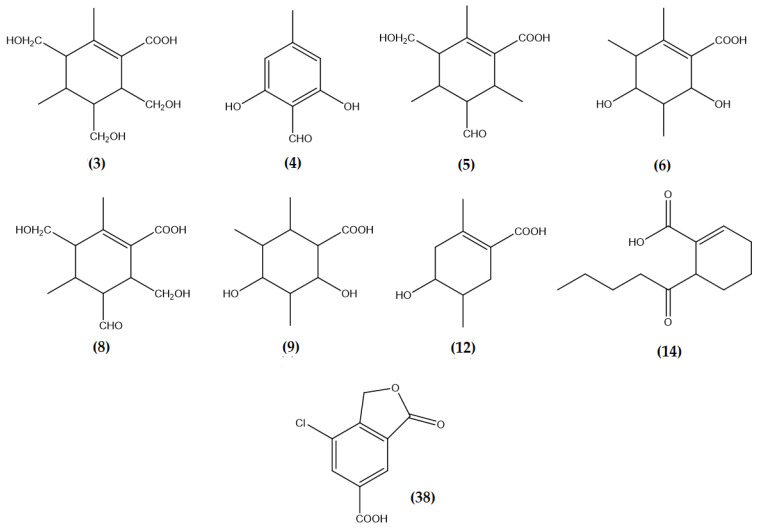
Chemical structure of Compound **3**, **4**, **5**, **6**, **8**, **9**, **12**, **14**, and **38** identified in the ME-hex of *Physcia mediterranea* by UHPLC/ESI/MS/MS.

**Figure 5 molecules-26-01121-f005:**
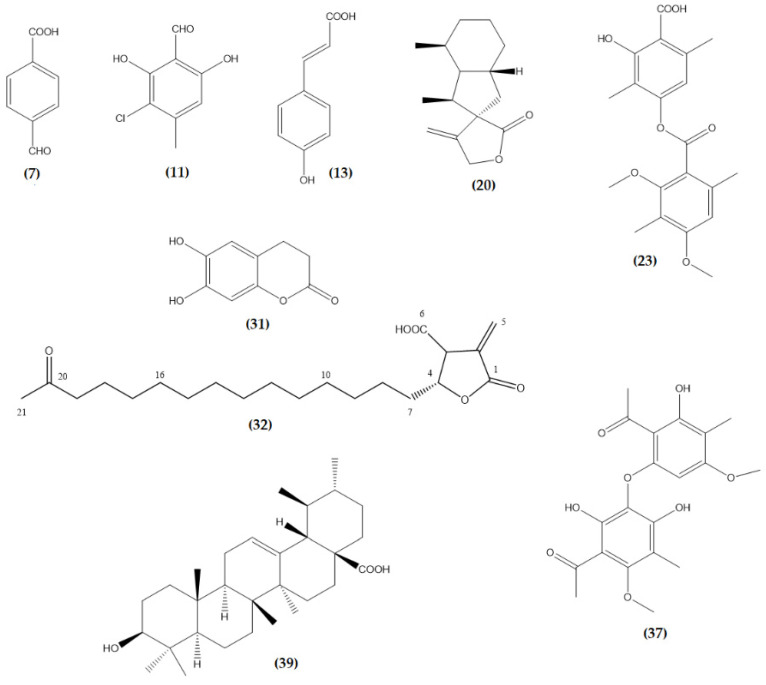
Chemical structure of compound **7**, **11**, **13**, **20**, **23**, **31**, **32**, **37**, and **39** identified in ME-hex of *Physcia mediterranea* by UHPLC/ESI/MS/MS.

**Figure 6 molecules-26-01121-f006:**
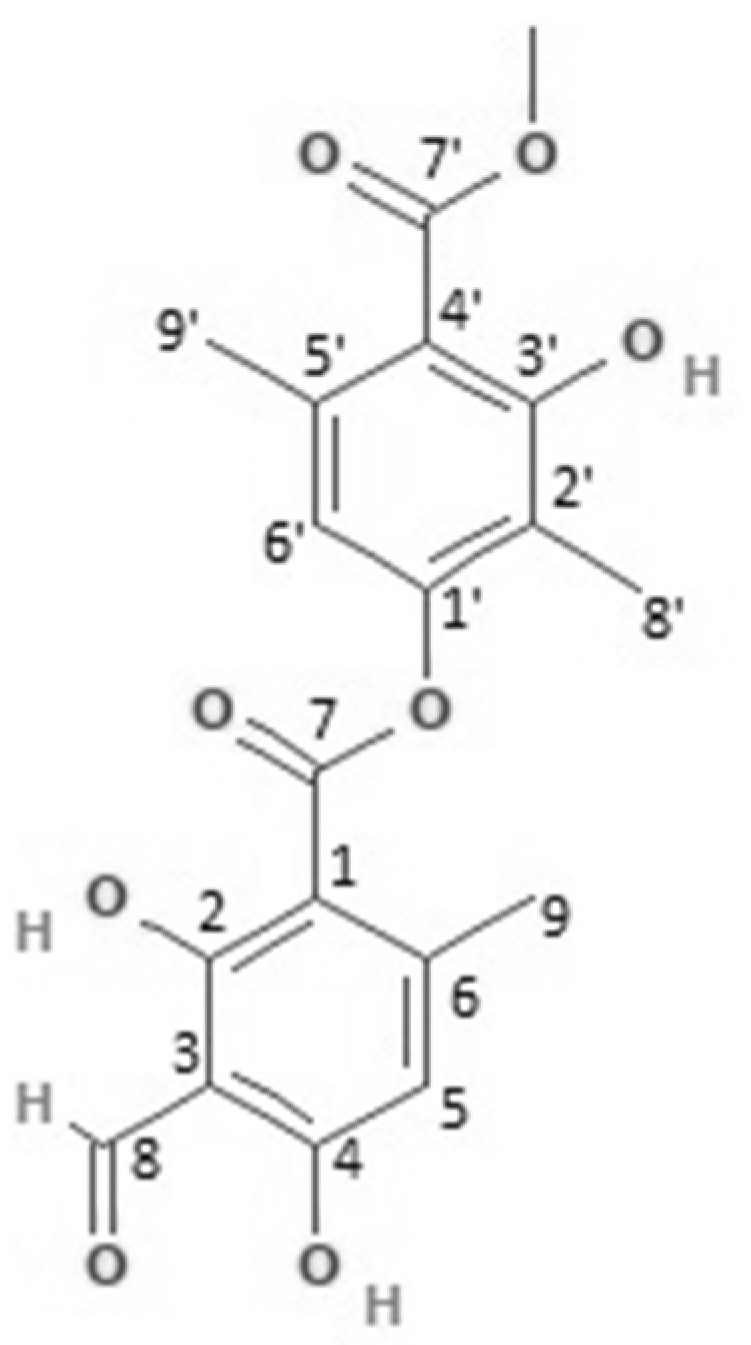
Chemical structure of atranorin isolated from WA-hex of *Physcia mediterranea.*

**Figure 7 molecules-26-01121-f007:**
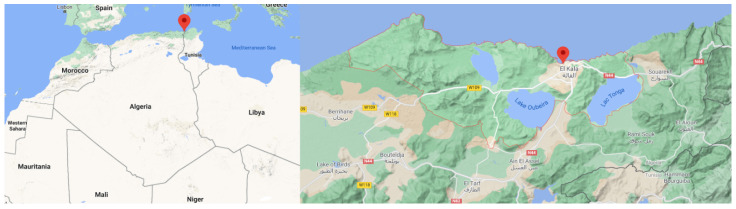
The topography of Algeria, indicating El Kala region boundaries (36°49′ 09″ N; 08°31′ 33″ E).

**Table 1 molecules-26-01121-t001:** Identification of metabolites in ME-hex of *Physcia mediterranea* by GC-MS analysis.

N°	RT	Compound	Mass	%	Synonyms	
1	19:56	Methyl 2-hydroxy-4-methoxy-6-methylbenzoate	196	0.3	Orsellinic Acid Methyl Ester 4-Methyl Ether (Sparassol)	C_10_H_12_O_4_
2	20:01	Methyl 2,4-dihydroxy-3,5,6-trimethylbenzoate	210	52.3	Derivatives of methylbenzoic acids	C_11_H_14_O_4_
3	20:43	Methyl 4-hydroxy-2-methoxy-3,6-dimethylbenzoate	210	1.1	2-metoxy-Atraric acid	C_10_H_12_O_4_
4	20:64	Methyl 2,4-dihydroxy-3,6-dimethylbenzoate	196	2.3	Atraric acid	C_10_H_12_O_4_
5	22:55	Methyl hexadecanoate	270	24.2	Palmitic acid	C_16_H_32_O_2_
6	23:53	13-Methyl-17-norkaur-15-ene	272	2.4	Linoleic acid	C_18_H_32_O_2_
7	24:19	Methyl *cis*, *cis*-9,12-octadecadienoate	294	3.2	Oleic acid	C_18_H_34_O_2_
8	24:25	Methyl (*Z*)-9-octadecenoate	296	4.9	Stearic acid	C_18_H_34_O_2_
9	24:48	Methyloctadecanoate	298	3.8	(−)-ent-Kauran-16α-ol	C_2O_H_34_O
10	26:21	Unknown	376	2.2	Unknown	
T%				95.5		

RT = retention time; T% = total of compounds identified (%) in the extracts.

**Table 2 molecules-26-01121-t002:** Identification of metabolites in *WA-hex* of *Physcia mediterranea* by UHPLC/ESI/MS/MS.

N°	RT	[M − H]^−^	TM	MM	MT	Compounds Identification
1	0.03	146.9397	147.9475	C_4_H_4_O_6_	A	Dihydroxyfumaric acid
2	1.80	116.9276	117.9354	C_4_H_6_O_4_	A	Butendioic acid
3	2.72	187.0971	188.1049	C_9_H_16_O_2_	A	3,5-dimethoxyciclohexanecarboxylic acid
4	5.51	163.0395	164.0473	C_9_ H_8_ O_3_	B	*p*-Coumaric acid
5	6.57	227.1286	228.1364	C_12_H_20_O_4_	A	Trans-dodec-2-enedioic acid (traumatic acid)
6	7.46	282.2077	283.2155	C_16_ H_29_ NO_3_	A	*N*-dodecanoyl-l-Homoserine lactone
7	8.09	174.9556	175.9634	C_5_ H_4_ O_7_	A	2-Hydroxy-3,4-dioxopentanedioc acid
8	8.15	293.1762	294.1840	C_17_H_26_O_4_	B	Gingerol
9	14.37	295.2280	296.2358	C_18_ H_32_ O_3_	A	18-Hydroxylinoleic acid
10	15.36	177.0186	178.0264	C_9_ H_6_ O_4_	B	6,7-Dihydroxycoumarin (esculetin)
11	15.44	373.0929	374.1007	C_19_ H_18_ O_8_	C	Atranorin
12	16.47	407.0539	408.0617	C_19_ H_17_ O_8_Cl	C	Chloroatranorin
13	17.57	277.2175	278.2203	C_18_H_30_O_2_	A	Octadeca-9,12,15-trienoic acid (linolenelaidic acid)
14	17.77	265.1480	266.1558	C_15_ H_22_ O_4_	A	(4*E*,6*E*,9*E*)-Pentadeca-4,6,9-trienedioic acid
15	18.03	253.2331	254.2249	C_16_ H_30_ O_2_	A	Palmitoleic acid
16	18.34	241.2173	242.2251	C_15_H_30_O_2_	A	Pentadecanoic acid
17	18.50	279.0936	280.2409	C_18_ H_32_ O_2_	A	Linoleic acid
18	18.91	267.2331	268.2409	C_17_ H_32_ O_2_	A	*cis*-9-Heptadecenoic acid (margaroleic acid)
19	19.15	255.2329	256.2407	C_16_ H_32_ O_2_	A	Palmitic acid
20	19.35	459.3271	460.3349	C_25_H_48_O_7_	Unknown	Unknown
21	19.51	281.2487	282.2565	C_18_ H_34_ O_2_	A	Oleic acid
22	19.75	459.3272	460.3350	C_25_ H_48_ O_7_	A	Methyl glucose isostearate
23	19.89	269.2488	270.2566	C_17_ H_34_ O_2_	A	Heptadecanoic acid (margaric acid)
24	19.99	307.2645	308.2723	C_20_ H_36_ O_2_	A	11,14-Eicosadienoic acid
25	20.07	457.3722	458.3800	C_27_ H_54_ O_5_	Unknown	Unknown
26	20.18	295.2645	296.2723	C_19_H_36_O_2_	A	10*E*-nonadecenoic acid
27	20.54	283.2643	284.2721	C_18_ H_36_ O_2_	A	Stearic acid (octadecanoic acid)
28	20.82	309.2800	310.2878	C_20_ H_38_ O_2_	A	Eicosenoic acid (gondoic acid)
29	21.71	311.2957	312.3035	C_20_H_40_O_2_	A	Arachidic acid (eicosanoic acid)
30	22.62	297.1532	298.1610	C_12_H_26_O_8_	Unknown	Unknown
31	22.77	339.3268	340.3346	C_22_H_44_O_2_	A	Docosanoic acid (behenic acid)
32	23.08	309.1743	310.1821	C_17_ H_26_ O_5_	D	Portentol
33	23.58	353.2003	354.2081	C_19_H_30_O_6_	Unknown	Unknown
34	23.81	311.1689	312.1767	C_13_ H_30_ O_8_	Unknown	Unknown
35	23.89	367.3579	368.3657	C_24_H_48_O_2_	A	Lignoceric acid (tetracosanoic acid)
36	24.01	397.2266	398.2344	C_21_H_34_O_7_	D	Stephanol
37	24.60	293.1793	294.1871	C_17_H_26_O_4_	A	Heptadecatrienedioic acid
38	24.94	325.1844	326.1922	C_14_ H_30_ O_8_	Unknown	Unknown
39	25.33	395.3895	396.3973	C_26_H_52_O_2_	A	Hexacosanoic acid or cerotic acid
40	25.89	337.2055	338.2133	C_19_ H_30_ O_5_	A	6-Oxononadeca-8,11-dienedioic acid
41	26.10	339.2000	340.2078	C_15_ H_32_ O_8_	Unknown	Unknown
42	26.24	381.2317	382.2395	C_21_ H_34_ O_6_	A	19-Acetoxylichesterinic acid
43	26.75	425.2581	426.2659	C_23_H_38_O_7_	E	Asebotoxin I
44	27.53	321.2106	322.2184	C_19_H_30_O_4_	A	Nonadecatrienedioic acid
45	27.85	304.9143	305.9221	Unknown	Unknown	Unknown

RT = retention time; TM = theoretical mass (*m*/*z*); [M − H]^−^ = measured mass as negative ion (*m*/*z*); MM = molecular mass; MT = metabolite type; A = paraconic and aliphatic acids; B = phenolic compounds; C = depsides; D = cycloaliphatic compounds; E = diterpenes.

**Table 3 molecules-26-01121-t003:** Identification of metabolites in ME-hex of *Physcia mediterranea* by UHPLC/ESI/MS/MS.

N°	RT	[M – H]^−^	TM	MM	MT	Compounds	MS^2^ Ions
1	2.72	187.0970	188.1043	C_9_H_16_O_4_	A	3,5-Dimethoxyciclohexanecarboxylic acid	141.8670; 123.0807
2	2.75	293.0669	294.0741	C_14_H_14_O_7_	A	6-(Hydroxymethyl)-3,5-bis(methoxycarbonyl)-2,4-dimethylcyclohex-1-ene-1-carboxylic acid	234.0530; 207.0695
3	3.08	243.1239	244.1311	C_12_H_20_O_5_	A	3,5,6-Hydroxymethyl-2,4-dimethylcyclohex-1-ene-1-carboxylic acid	183.1025; 176.6474
4	3.15	151.0393	152.0465	C_8_H_8_O_3_	B	Atranol	123.0444; 105.0335; 81.0335
5	3.20	225.1129	226.1201	C_12_H_18_O_4_	A	5-Formyl-3-hydroxymethyl-2,4,6-trimethylcyclohex-1-ene-1-carboxylic acid	207.1023; 97.0284
6	3.47	199.0973	200.1046	C_10_H_16_O_4_	A	3,5-Dihydroxy-2,4,6-trimethylciclohexenecarboxilic acid	183.4492; 162.8385; 114.9508
7	3.58	149.0237	150.0310	C_8_H_6_O_3_	C	4-Formylbenzoic acid	−
8	3.82	241.1081	242.1153	C_12_H_18_O_5_	A	5-Formyl-3,6-dihydroxymethyl-2,4-dimethylcyclohex-1-enecarboxylic acid	218.816; 172.6485
9	4.10	201.1129	202.1202	C_10_H_18_O_4_	A	2,4-Dihydroxy-3,5,6-trimethylcyclohexane-1-carboxylic acid	197.6263; 164.8350; 139.1124
10	4.95	199.1337	200.1409	C_11_H_20_O_3_	A	2-Hydroxy-10-undecenoic acid	−
11	5.00	185.0006	186.0079	C_8_H_7_ClO_3_	B	Chloroatranol	−
12	5.21	169.0863	170.0936	C_9_H_14_O_3_	A	4-Hydroxy-2,5-dimethylcyclohex-1-ene-1-carboxylic acid	124.9796; 104.4180
13	5.49	163.0392	164.0470	C_9_H_8_O_3_	D	*p*-Coumaric acid	−
14	5.60	209.1181	210.1253	C_12_H_18_O_3_	A	6-(1-Oxopentyl)-1-cyclohexene-1-carboxylic acid	146.9850; 105.0336
15	5.87	215.1286	216.1359	C_11_H_20_O_4_	A	Undecanedioic acid	−
16	6.57	227.1288	228.1360	C_12_H_20_O_4_	A	trans-Dodec-2-enedioic acid	−
17	7.44	282.2078	283.2150	C_16_H_29_NO_3_	A	*N*-Dodecanoyl-l-Homoserine lactone	−
18	9.53	243.1601	244.1675	C_13_H_24_O_4_	A	Tridecanedioic acid	−
19	10.59	311.2230	312.2302	C_18_H_32_O_4_	A	9*Z*-Octadecenedioic acid	−
20	10.87	233.1547	234.1619	C_15_H_22_O_2_	E	Fukinanolide	−
21	10.89	323.2230	324.2303	C_19_H_32_O_4_	F	allo-Protolichesterinic acid	−
22	12.28	313,2388	314.2266	C_18_H_34_O_4_	A	Octadecanedioic acid	−
23	13.04	389.1245	390.1315	C_20_H_22_O_8_	B	8-Hydroxydiffractaic acid	−
24	13.34	293.2124	294.2202	C_18_H_30_O_3_	A	2-Hydroxylinolenic acid	−
25	13.58	291.1968	292.2041	C_18_H_28_O_3_	A	α-Licanic acid	−
26	14.38	295.2279	296.2351	C_18_H_32_O_3_	A	2-Hydroxylinoleic acid	−
27	14.40	295,2278	296,2352	C_18_H_32_O_3_	A	18-hydroxylinoleic acid	−
28	15.19	297.2436	298.2508	C_18_H_34_O_3_	A	9-Oxooctadecanoic acid	−
29	15.25	295.2280	296.2351	C_18_H_32_O_3_	A	Coriolic acid	−
30	15.36	297.2435	298.2508	C_18_H_34_O_3_	A	Ricinoleic acid	−
31	15.43	177.0187	178.0259	C_9_H_6_O_4_	D	6,7-Dihydroxycoumarin	−
32	15.46	365.2330	366.2403	C_21_H_34_O_5_	G	Muronic acid	−
33	16.37	471.3481	472.3553	C_30_H_48_O_4_	Unknown	Unknown	−
34	16.47	210.9834	211.9873	C_9_H_5_ClO_4_	H	7-chloro-3-oxo-1,3-dihydroisobenzofuran-5-carboxylic acid	138.9948, 103.0180
35	16.51	407.0540	408.0611	C_19_H_17_ClO_8_	B	Chloroatranorin	−
36	17.59	387.2544	388.2616	C_24_H_36_O_4_	Unknown	Unknown	−
37	17.68	389.1242	390.1314	C_20_H_22_O_8_	I	Leprolomin	−
38	18.04	253.2173	254.2244	C_16_H_30_O_2_	A	Palmitoleic acid	−
39	18.33	241.2172	242.2245	C_15_H_30_O_2_	A	Pentadecanoic acid	−
40	18.35	455.3531	456.3605	C_30_H_48_O_3_	G	Ursolic acid	−
41	18.50	279.2330	280.2403	C_18_H_32_O_2_	A	Linoleic acid	−
42	18.66	299.2595	300.2667	C_18_H_36_O_3_	A	2-Hydroxyoctadecanoic acid	−
43	18.78	279.2332	280.2403	C_18_H_32_O_2_	A	Linoleic acid	−
44	19.16	255.2329	256.2401	C_16_H_32_O_2_	A	Palmitic acid	−
45	19.49	281.2485	282.2559	C_18_H_34_O_2_	A	Oleic acid	−
46	19.62	269.2488	270.2561	C_17_H_34_O_2_	A	15-Methylhexadecanoic acid	−
47	19.89	269.2488	270.2561	C_17_H_34_O_2_	A	Heptadecanoic acid	−
48	20.07	327.2543	328.2616	C_20_H_40_O_3_	A	2-Hydroxyeicosanoic acid	−
49	20.54	283.2643	284.2716	C_18_H_36_O_2_	A	Stearic acid (octadecanoic acid)	−
50	20.74	309.2801	310.2875	C_20_H_38_O_2_	A	Eicosenoic acid (Gondoic acid)	−
51	21.14	297.2801	298.2873	C_19_H_38_O_2_	A	Nonadecanoic acid	−
52	21.70	311.2957	312.3029	C_20_H_40_O_2_	A	Arachidic acid (Eicosanoic acid)	−
53	22.68	637.4841	638.4908	C_41_ H_66_ O_5_	Unknown	Unknown	−
54	24.51	605.3483	606. 3561	C_37_H_50_O_7_	Unknown	Unknown	−

RT = retention time (min); TM = theoretical mass (*m*/*z*); [M – H]^−^ = measured mass as negative ion (*m*/*z*); MM = molecular mass; MT = metabolite type; A = paraconic and aliphatic acids; B = depsides; C = carboxybenzaldehyde; D = phenolic acids; E = sesquiterpene lactones; F = depsones; G = triterpene; H = carboxyphthalide; I = diphenyl ether.

## Data Availability

Lichen of *Physcia mediterranea* Nimis was used in this study. Professor Monia Ali Ahmed identified *Physcia mediterranea* Nimis, lichenologist and research director of the Pathology of Ecosystems team at the University of Badji-Mokhtar, Annaba, Algeria. The identification was confirmed by Pr Jean Michel Sussey, lichenologist at the French Association of Lichenology (AFL). This sample has been deposited in Badji-Mokhtar University, Annaba, code AAM-1.

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
