# Peer review of "Phytochemical Investigation of New Algerian Lichen Species: Physcia Mediterranea Nimis"

_molecules, 2021, doi:10.3390/molecules26041121_

Round 1

Reviewer 1 Report

I'm sorry but the new version of the manuscript is lacking from a methodological and scientific point of view. There are many errors and inaccuracies and here and here I list some of them.

1) The authors state in the title of biological activity of the studied lichens, but unfortunately no biological tests on cell lines have been reported.

2) although the analytical procedures used are valid, they do not complete the analysis of all the components of the plant species studied.

3) A scientific work MUST have a hypothesis that with scientific results (statistically valid) is demonstrated and discussed. This approach is missing from this manuscript. The paper is descriptive only. a serious statistical survey is totally absent.

Author Response

Response 1: Thank you for your comment.

The term "Bioactive Compounds" was used in the title not to refer to biological tests (in vitro or in vivo) but to the presence of these compounds in the lichen that has never been studied before (“New source”). This is because "Bioactive Compounds" are defined as compounds present in small amounts in foods, mainly in fruits, vegetables and whole grains, and provide health benefits beyond the basic nutritional value (Gökmen, 2016). Currently, many bioactive compounds are considered as excellent alternative candidates to synthetic antioxidants or antimicrobials food additives possessing non side effects in comparison with synthetic additives (Hamzalıoğlu et al., 2016). They are molecules that may have therapeutic potential by influencing energy intake, while reducing pro-inflammatory conditions, oxidative stress and metabolic disorders (Siriwardhana et al., 2013). To date, many bioactive compounds have been discovered. The structure and chemical function of these compounds are highly variable, and they are grouped accordingly (Mocan et al., 2018). Bioactive compounds are extracted from natural sources by solid-liquid extraction using organic solvents and other techniques, such as supercritical fluid extraction, high pressure processes, microwave-assisted extraction, subcritical water extraction, and ultrasound-assisted extraction (Khezerlou et al., 2020).

In our study we used solid-liquid extraction using organic solvents and overall result obtained suggests that lichens are also a source of bioactive compounds whose different biological activities have been demonstrated by numerous investigations (this has been extensively described in the discussion section for each identified compound, please see l.126-372 in the revised manuscript). In recent years, there has been a new trend worldwide in the search for new safe phytochemicals for the pharmaceutical or nutraceutical industries (Magnuson et al., 2013). This trend corresponds to a strong consumer demand for safe and high-quality food phytochemicals, which can be attributed in part to the widespread availability and accessibility of quality health data and information (Tajkarimi et al., 2010). According to our study, the presence of phytochemicals and other bioactive compounds in lichen could serve as a potential new source of medicines in the future.

References:

Gökmen, V. (2016). Acrylamide in food: Analysis, content and potential health effects. Elsevier.

Andrei Mocan, Gokhan Zengin, Adriano Mollica, Ahmet Uysal, Erdogan Gunes, Gianina Crişan, Abdurrahman Aktumsek, Biological effects and chemical characterization of Iris schachtii Markgr. extracts: A new source of bioactive constituents, Food and Chemical Toxicology, Volume 112, 2018, Pages 448-457, https://doi.org/10.1016/j.fct.2017.08.004.

Siriwardhana, N., Kalupahana, N. S., Cekanova, M., LeMieux, M., Greer, B., & MoustaidMoussa, N. (2013). Modulation of adipose tissue inflammation by bioactive food compounds. Journal of Nutritional Biochemistry, 24(4), 613–623. https://doi.org/10.1016/j.jnutbio.2012.12.013

Aytül Hamzalıoğlu, Vural Gökmen, Chapter 18 - Interaction between Bioactive Carbonyl Compounds and Asparagine and Impact on Acrylamide, Editor(s): Vural Gökmen, Acrylamide in Food, Academic Press, 2016, Pages 355-376, https://doi.org/10.1016/B978-0-12-802832-2.00018-8.

Arezou Khezerlou, Seid Mahdi Jafari, 13 - Nanoencapsulated bioactive components for active food packaging, Editor(s): Seid Mahdi Jafari, Handbook of Food Nanotechnology, Academic Press, 2020, Pages 493-532, https://doi.org/10.1016/B978-0-12-815866-1.00013-3.

  1. Magnuson, I. Munro, P. Abbot, N. Baldwin, R. Lopez-Garcia, K. Ly, L. McGirr, A. Roberts, S. Socolovsky Review of the regulation and safety assessment of food substances in various countries and jurisdictions Food Addit. Contam. Part A. Chem. Anal. Control. Expo. Risk Assess., 30 (2013), pp. 1147-1220, 10.1080/19440049.2013.795293

Close M.M. Tajkarimi, S.A. Ibrahim, D.O. Cliver Antimicrobial herb and spice compounds in food. Food Control, 21 (2010), pp. 1199-1218, 10.1016/j.foodcont.2010.02.003

Response 2: Thank you for your comment.

Our work is a preliminary study on a lichen that has never been studied before. There are no studies on the phytochemical characterization of this species. The results encourage us to extend our work on this lichen and its products through testing them on in vitro and in vivo tests on experimental model systems.

Response 3: A phytochemical analysis of a species that has never been studied before is also a scientific work. The chemical analysis of a material, whether it is known in folk medicine or observed as part of a planned screening programme, is the starting point for any research into potential drugs.

Our work is not a descriptive study. Through our analysis we wanted to demonstrate that lichens are valuable plant resources and are used as medicines, food, fodder, perfumes, spices, dyes and for various purposes all over the world. Compounds isolated from various lichen species have been reported to have diverse biological activities (Shukla et al.,2010; Shrestha et al., 2013).

Several works like ours have also been published without any statistical analysis (Oniszczuk, 2016; Salgado et al., 2018; Pascale et al.,2020).

References:

Shukla, V., Joshi, G.P. & Rawat, M.S.M. Lichens as a potential natural source of bioactive compounds: a review. Phytochem Rev 9, 303–314 (2010). https://doi.org/10.1007/s11101-010-9189-6

Shrestha, G., St. Clair, L.L. Lichens: a promising source of antibiotic and anticancer drugs. Phytochem Rev 12, 229–244 (2013). https://doi.org/10.1007/s11101-013-9283-7

Oniszczuk, A. LC-ESI-MS/MS Analysis and Extraction Method of Phenolic Acids from Gluten-Free Precooked Buckwheat Pasta. Food Anal. Methods 9, 3063–3068 (2016). https://doi.org/10.1007/s12161-016-0489-3

Salgado, F., Albornoz, L., Cortéz, C., Stashenko, E., Urrea-Vallejo, K., Nagles, E., Galicia-Virviescas, C., Cornejo, A., Ardiles, A., Simirgiotis, M., García-Beltrán, O., & Areche, C. (2017). Secondary Metabolite Profiling of Species of the Genus Usnea by UHPLC-ESI-OT-MS-MS. Molecules (Basel, Switzerland), 23(1), 54. https://doi.org/10.3390/molecules23010054

Pascale, R., Acquavia, M.A., Cataldi, T.R.I. et al. Profiling of quercetin glycosides and acyl glycosides in sun-dried peperoni di Senise peppers (Capsicum annuum L.) by a combination of LC-ESI(-)-MS/MS and polarity prediction in reversed-phase separations. Anal Bioanal Chem 412, 3005–3015 (2020). https://doi.org/10.1007/s00216-020-02547-2

Reviewer 2 Report

The manuscript “algerian licen Physcia mediterránea Nimis: a new source of bioactive compounds” is an interesting document due to there is very few information of the chemical composition of this product reported, nevertheless there are comments that need to be addressed prior publication. The title of the manuscript presented a Physcia mediterranea as a source of bioactive compounds, nevertheless in results and discussion sections there is not information of bioactive analysis, information of bioactivity must be added, or the title must be changed to chemical composition of…. In the abstract section, the type of NMR used is not indicated and the conclusion is so ambiguous because there is not information presented of the bioactivity. In introduction section authors presented wide information of the importance of the lichen, nevertheless there is not information of the importance of the bioactive compounds and extraction conditions. In results and discussion section, the quality of the figure 1 must be improved to observe clearly the explanation presented in the text. In figure 2 the chemical structures must be homogenized. Figure 3 and figure 4 must be added to supplementary data. The information presented in tables 2 and 3 described the information of the figures 3 and 4. To a better understanding, a better classification of the compounds obtained must be included in the discussion and tables. Figures 9 and 8 must be added to supplementary data and the summary of the results and the discussion of them must be added in the text. Figure 10 present in material and methods section could be changed to results and discussion section, or corrected the number of figure. Why do the authors used soxhlet extraction?, There is good information about the chemical composition of the product but the title and the introduction section is not in concordance with the results.

Author Response

Response 1: Thank you for your comment.

The term "Bioactive Compounds" was used in the title not to refer to biological tests (in vitro or in vivo) but to the presence of these compounds in the lichen that has never been studied before (“New source”). This is because "Bioactive Compounds" are defined as compounds present in small amounts in foods, mainly in fruits, vegetables and whole grains, and provide health benefits beyond the basic nutritional value (Gökmen, 2016). Currently, many bioactive compounds are considered as excellent alternative candidates to synthetic antioxidants or antimicrobials food additives possessing non side effects in comparison with synthetic additives (Hamzalıoğlu et al., 2016). They are molecules that may have therapeutic potential by influencing energy intake, while reducing pro-inflammatory conditions, oxidative stress and metabolic disorders (Siriwardhana et al., 2013). To date, many bioactive compounds have been discovered. The structure and chemical function of these compounds are highly variable, and they are grouped accordingly (Mocan et al., 2018).

In our study we used solid-liquid extraction using organic solvents and overall result obtained suggests that lichens are also a source of bioactive compounds whose different biological activities have been demonstrated by numerous investigations (this has been extensively described in the discussion section for each identified compound, please see l.126-372 in the revised manuscript). In recent years, there has been a new trend worldwide in the search for new safe phytochemicals for the pharmaceutical or nutraceutical industries (Magnuson et al., 2013). This trend corresponds to a strong consumer demand for safe and high-quality food phytochemicals, which can be attributed in part to the widespread availability and accessibility of quality health data and information (Tajkarimi et al., 2010). According to our study, the presence of phytochemicals and other bioactive compounds in lichen could serve as a potential new source of medicines in the future.

References:

Gökmen, V. (2016). Acrylamide in food: Analysis, content and potential health effects. Elsevier.

Andrei Mocan, Gokhan Zengin, Adriano Mollica, Ahmet Uysal, Erdogan Gunes, Gianina Crişan, Abdurrahman Aktumsek, Biological effects and chemical characterization of Iris schachtii Markgr. extracts: A new source of bioactive constituents, Food and Chemical Toxicology, Volume 112, 2018, Pages 448-457, https://doi.org/10.1016/j.fct.2017.08.004.

Siriwardhana, N., Kalupahana, N. S., Cekanova, M., LeMieux, M., Greer, B., & MoustaidMoussa, N. (2013). Modulation of adipose tissue inflammation by bioactive food compounds. Journal of Nutritional Biochemistry, 24(4), 613–623. https://doi.org/10.1016/j.jnutbio.2012.12.013

Aytül Hamzalıoğlu, Vural Gökmen, Chapter 18 - Interaction between Bioactive Carbonyl Compounds and Asparagine and Impact on Acrylamide, Editor(s): Vural Gökmen, Acrylamide in Food, Academic Press, 2016, Pages 355-376, https://doi.org/10.1016/B978-0-12-802832-2.00018-8.

Arezou Khezerlou, Seid Mahdi Jafari, 13 - Nanoencapsulated bioactive components for active food packaging, Editor(s): Seid Mahdi Jafari, Handbook of Food Nanotechnology, Academic Press, 2020, Pages 493-532, https://doi.org/10.1016/B978-0-12-815866-1.00013-3.

  1. Magnuson, I. Munro, P. Abbot, N. Baldwin, R. Lopez-Garcia, K. Ly, L. McGirr, A. Roberts, S. Socolovsky Review of the regulation and safety assessment of food substances in various countries and jurisdictions Food Addit. Contam. Part A. Chem. Anal. Control. Expo. Risk Assess., 30 (2013), pp. 1147-1220, 10.1080/19440049.2013.795293

Close M.M. Tajkarimi, S.A. Ibrahim, D.O. Cliver Antimicrobial herb and spice compounds in food. Food Control, 21 (2010), pp. 1199-1218, 10.1016/j.foodcont.2010.02.003

Response 2: Thank you for your comment.

The information has been added to the abstract. Please see the l.25 of the revised manuscript.

Response 3: Thank you for your comment.

As described in point 1 above, we have used the term "bioactive compounds" to indicate and disclose the importance of lichens as a promising source of these compounds, and this has been detailed in all parts of the manuscript including the conclusion. Knowledge of the chemical constituents of lichens is invaluable as this information will be useful for the synthesis of potential new chemicals that could play a key role in human health.

Response 4: Thank you for your comment.

The information has been added to the introduction. Please see the l.73 to l.83 of the revised manuscript

Response 5: Thank you for your observation.

Comparing the results of the Table 1 with the Figure 2 you will notice that according to the retention time some spades are getting closer and are combined into one peak, as for the peak of 1, 3, and 4 with the main peak 2. We have tried to explain the Figure 1 based on the results of the Table 1, i.e. the percentage of each identified compound. Thus, in the text it is mentioned that out of the 95.5% of the compounds identified there were (56%) derivatives of methylbenzoic acids and (36.1%) free fatty acids. A diterpene with a Kauran skeleton ((-)-ent-Kauran-16a-ol) (3.8%) was also identified. and only 2.2% of the whole extract was not identified. (please see l.109-120 of the revised manuscript).

Response 6: Thank you for your comment.

In the Figure 3, the chemical structures are corrected. (please see 137.).

The Figure 4 and Figure 5 were added to the Supplementary data S1 and S2. (Please see l.169, l.174 and l.176 of revised manuscript).

Response 7: Thank you for your comment.

All changes have been made to the manuscript. The changes in the inverted figures have imported (Please see Figure 6 and in Supplementary data S1). The classification of compounds is also changed in the tables and discussion. (Please see Table 2 and 3, l.179, l.203, 213-217, l.240, l.282-320 in the revised manuscript).

Response 8: Thank you for your comment. The Figure 10 and 11 have been added to the Supplementary data S3. The results were discussed (please see l.338-372).

Response 9: Thank you for your comment. The figure (1) has been moved to the introduction part and in its place, we have added figure 11 which represents the lichen harvesting area. (Please see l.108 of revised manuscript).

Response 10: Thank you for your comment.

As said in the discussion part Soxhlet extraction method is the best choice in studying organic analytes extracted from lichens (Forbes et al., 2020). It is still used to date to extract organic air pollutants, organochlorinated pesticides and insecticides from the lichen matrix (Augusto et al., 2004; Van der Wat and Fobes), 2019. Additionally, the separation of the polar compounds, i.e. the fatty acids, has been well detailed in section 2.1. As it has been well explained in the text, the determination of the lipid profile (chemotype) is crucial for the taxonomic identification of lichen species (Please see l.139-168). During the extraction process, several compounds can be detected such as Depsides, Depsones, phenolic compounds, terpenes and others (Shukla et al., 2010).

References:

Forbes, P.; Wat, L.; Strumpher, J. Comparative Perspectives on Extraction Methods for Organic Metabolites and Pollutants from Lichens. In Lichen‐Derived Products; Wiley, 2020; pp. 27–73.

Augusto, S.; Pinho, P.; Branquinho, C.; Pereira, M.J.; Soares, A.; Catarino, F. Atmospheric Dioxin and Furan Deposition in Relation to Land-Use and Other Pollutants: A Survey with Lichens; Kluwer Academic Publishers, 2004; Vol. 49;.

Van der Wat, L.; Forbes, P.B.C. Comparison of extraction techniques for polycyclic aromatic hydrocarbons from lichen biomonitors. Environ. Sci. Pollut. Res. 2019, 26, 11179–11190, doi:10.1007/s11356-019-04587-3.

Shukla, V., Joshi, G.P. & Rawat, M.S.M. Lichens as a potential natural source of bioactive compounds: a review. Phytochem Rev 9, 303–314 (2010). https://doi.org/10.1007/s11101-010-9189-6

Response 11: Thank you for your comment.

This has been discussed in point 1.

Round 2

Reviewer 1 Report

Dear Authors,

I thank you for your long replies to my objections, which unfortunately do not resolve the strong critical issues that still exist.

In my opinion, the title should be changed to “Chemical characterization of Algerian lichen Physcia mediterranea Nimis”. The characterization of bioactive molecules should always be accompanied by a study on biological matrices. Indeed, it is not enough to say that a molecule is antioxidant, but it is necessary to study it in the context in which the molecule is synthesized. I suggest the following manuscript to the authors to be included in their references and which could help them clarify some aspects of biological activity: Gupta AK et al. Artocarpus lakoocha Roxb. and Artocarpus heterophyllus Lam. Flowers: New Sources of Bioactive Compounds. Plants (Basel). 2020 Oct 9;9(10):1329. doi: 10.3390/plants9101329.

Authors should discuss the concept of PHYTOCOMPLEX more. Above all, in the characterization of new plant species it is important to evaluate the endemic characteristics. As the authors surely know, the same plant species that grow in different ecosystems produce metabolites in different concentrations. It follows that the antioxidant power is different.

These aspects have not been evaluated by the authors.

Another aspect that impoverishes the manuscript is the specificity of the solvents used. If the extracts of a plant species will be used as food supplements, the solvent used for the extraction should also be compatible for food uses.

Finally, the authors state: “According to our study, the presence of phytochemicals and other bioactive compounds in lichen could serve as a potential new source of medicines in the future”, this type of statement, in my opinion, is very serious for the implications it could have in the ecological field. No researcher can state that a plant species linked to its ecological context can become a future source of medicine. Many ecological disasters originate from the reckless exploitation of already very fragile ecosystems. Furthermore, in order to be a true source of drugs it would have to undergo multiple preclinical and clinical tests.

For all these reasons, in my opinion the manuscript still suffers from many critical issues.

Author Response

Dear Reviewer 1,

Regarding the manuscript's title, I agree with your comment, and we can change it to this title: "Phytochemical study of new species of Algerian lichens: Physcia mediterranea Nimis".

Your proposal to evaluate bioactive molecules is a second step in isolating, purifying, and studying biological activities. In this case, in the first manuscript, it was indicated that a general evaluation of antioxidant activity was carried out, obtaining high antioxidant power as it was confirmed by the bibliographical references that were found for atranorine and chloroatranorine. As other research groups had already studied these activities, there was no need to repeat the work.

I want to inform you that the chemical compounds in this work have not been synthesized: they have been isolated and purified from a natural source such as lichen.

We have a lot of confusion with the comments you sent and the references to justify them. In our manuscript, we are doing a chemical study of the hexane extract of a lichen; its general metabolism and the secondary metabolites it produces are different from the vascular plants that you had indicated in your comment.

In this case, the biological activities of the products identified in this lichen refer to the data we were able to retrieve from the bibliography. These polyphenolic derivatives of lichens have nothing to do with the phenolic derivatives of the fruits, leaves, branches, and roots of plants of other countries' flora (vascular plants).

This is a general study on the extraction, separation and identification by different techniques and methods of the chemical components of a lichen extract.

I think that the term PHYTOCOMPLEX cannot be applied in this case, because we have a new species of lichen and we don't know at all whether it has other varieties and whether it exists in other countries. I think that in the case of lichens, the varieties of a species are more difficult because they come from a very complex association (symbiosis) between an alga, a bacteria or a fungus. Each one of them has its specificities. I agree that in the vascular plants of the flora of different countries, species variations occur more easily.

Logically, we cannot evaluate these factors because they do not exist, and what does not exist cannot be evaluated.

On the other hand, I want to explain how phytochemical studies are carried out: in these studies, we have always used (for several centuries) solvents that vary from the least polar: Hexane; those of medium polarity: chloroform or acetone; to those of higher polarity and which can be methanol or ethanol and finally we use the solvent with water of higher polarity. This proposal is general for vascular plants of the flora, algae, microalgae, lichens and other species.  For example, water or ethanol is used to extract dyes used in fabric cleaners, food dyes and others.

The extraction of plant material with hexane can be cold with a lower extraction power or hot at the boiling temperature of the hexane with a higher extraction power, this being the method that has worked best for lichen in this work. Logically, if we make an extraction of a vegetable matter for food use, it must be extracted with water and it can be cold, at 40 - 50º C and at the boiling temperature of the water, we can make an infusion as in the case of coffee and tea. In our work and in this manuscript, this has nothing to do with or is not related to a food use, for the moment.

Regarding the sentence you chose in our manuscript: "According to our study, the presence of phytochemicals and other bioactive compounds in lichens could be a potential new source of medicines in the future" and the comment you made, I think it is totally disconnected from reality. Since the last century, studies on flora, endemic plants, rare plants have been carried out without creating an ecological problem and I have worked with important botanists all over the world. The plant or lichen is studied, and if the plant is medicinal, aromatic, or to produce dyes or other products at an industrial level, the native flora is never used. Therefore, its large-scale use must be accompanied by controlled production in the field, in a greenhouse, in specific temperature, humidity, irrigation and nutrient systems. In the case of lichens, they are produced on a large scale to extract dyes for industrial use, certain essential oils, molecules and other derivatives for medical use.

In my university we apply this technique with the agricultural part and different species are adapted for its cultivation. I am really disappointed by this comment, which does not really apply to this work, if serious ecological problems have been created, it will not be because of this work.